# Exploring the Potential of *Crassostrea nippona* Hydrolysates as Dietary Supplements for Mitigating Dexamethasone-Induced Muscle Atrophy in C2C12 Cells

**DOI:** 10.3390/md22030113

**Published:** 2024-02-28

**Authors:** M. J. M. S. Kurera, D. P. Nagahawatta, N. M. Liyanage, H. H. A. C. K. Jayawardhana, D. S. Dissanayake, Hyo-Geun Lee, Young-Sang Kim, Sang In Kang, You-Jin Jeon

**Affiliations:** 1Department of Marine Life Sciences, Jeju National University, Jeju 63243, Republic of Korea; shehan.kurera@stu.jejunu.ac.kr (M.J.M.S.K.); pramuditha1992@jejunu.ac.kr (D.P.N.); liyanagenm@jejunu.ac.kr (N.M.L.); chathuri.k.j@stu.jejunu.ac.kr (H.H.A.C.K.J.); dinudissanayake95@stu.jejunu.ac.kr (D.S.D.); hyogeunlee92@jejunu.ac.kr (H.-G.L.); yskim@jejunu.ac.kr (Y.-S.K.); 2Department of Biotechnology, Faculty of Agriculture and Plantation Management, Wayamba University of Sri Lanka, Makandura, Gonawila 60170, NWP, Sri Lanka; 3Marine Science Institute, Jeju National University, Jeju 63333, Republic of Korea; 4Seafood Research Center, Silla University, Busan 49277, Republic of Korea

**Keywords:** *Crassostrea nippona*, enzyme hydrolysates, functional foods, marine shellfish, muscle atrophy, skeletal muscles

## Abstract

Muscle atrophy is a detrimental and injurious condition that leads to reduced skeletal muscle mass and disruption of protein metabolism. Oyster (*Crassostrea nippona*) is a famous and commonly consumed shellfish in East Asia and has become a popular dietary choice worldwide. The current investigation evaluated the efficacy of *C. nippona* against muscle atrophy, which has become a severe health issue. Mammalian skeletal muscles are primarily responsible for efficient metabolism, energy consumption, and body movements. The proteins that regulate muscle hypertrophy and atrophy are involved in muscle growth. *C. nippona* extracts were enzymatically hydrolyzed using alcalase (AOH), flavourzyme (FOH), and protamex (POH) to evaluate their efficacy in mitigating dexamethasone-induced muscle damage in C2C12 cells in vitro. AOH exhibited notable cell proliferative abilities, promoting dose-dependent myotube formation. These results were further solidified by protein expression analysis. Western blot and gene expression analysis via RT-qPCR demonstrated that AOH downregulated MuRF-1, Atrogin, Smad 2/3, and Foxo-3a, while upregulating myogenin, MyoD, myosin heavy chain expression, and mTOR, key components of the ubiquitin–proteasome and mTOR signaling pathways. Finally, this study suggests that AOH holds promise for alleviating dexamethasone-induced muscle atrophy in C2C12 cells in vitro, offering insights for developing functional foods targeting conditions akin to sarcopenia.

## 1. Introduction

Sarcopenia is a health condition that leads to diminished skeletal muscle mass, muscle fiber number, and muscle area with increasing age. It destructively impacts a healthy life and is a high-risk factor for the physical disability known as muscle atrophy. Muscle atrophy leads to an interruption in the balance between protein synthesis and degradation, lessening the protein content, reducing fiber diameter, and forming fatigue resistance in the body [1]. At ages above 60, the potential occurrence of muscle atrophy-related sarcopenia increases to approximately 5–13% [2]. Hence, these unfavorable conditions influence the regular lifestyle of humans and impair their overall well-being. Therefore, functional foods, pharmaceuticals, and therapeutic products must address this health condition. Muscle protein synthesis and breakdown are the two significant processes of skeletal muscle growth. When conditions exceed protein synthesis during protein breakdown, net protein synthesis occurs in the body, leading to muscle growth [3]. Exercise is a key factor in muscle protein metabolism. Exception for exercise, proper nutrition, and hormonal stimulation are highly impacted by muscle development [4]. Skeletal muscles consist of bundles of myocytes, each of which contains myofibrils that permit muscle contraction. When myofibrils heave, they increase muscle strength and density [5]. Skeletal muscles comprise approximately two-fifths of the total body mass and contribute to the convention of creating movements, maintaining body posture and position, regulating body temperature, accumulating nutrients, soothing joints, and maintaining overall metabolic homeostasis. Therefore, proper muscle conditions are necessary to maintain a healthy body [6]. 

Marine organisms are potential candidates for producing valuable secondary metabolites, especially bioactive compounds such as alkaloids, flavonoids, phenolic compounds, terpenes, and steroids. These compounds have antioxidant, anti-microbial, anti-diabetic, anti-hypertension, anti-obesity, anti-cancer, and anti-aging properties [7,8,9,10,11,12,13]. Further, due to their high potential, they contribute to drug discovery, cancer therapies, anti-microbial functions, and many other applications in the current world effectively [14,15,16,17,18,19,20]. Shellfish play a crucial role in supporting muscle hypertrophy due to their rich nutritional profile. These marine delicacies are abundant sources of high-quality proteins, essential amino acids, and bioactive compounds, all of which are fundamental for muscle growth and repair. Proteins in shellfish contribute to synthesizing muscle proteins, fostering the development and maintenance of lean muscle mass. Additionally, shellfish are notable for providing important micronutrients such as zinc, selenium, and vitamin B12, which play key roles in various metabolic processes, including those involved in muscle function and growth [21,22]. The omega-3 fatty acids found in certain shellfish varieties further contribute to muscle health by reducing inflammation and supporting overall muscle function. Including shellfish in the diet can be a valuable strategy for individuals aiming to enhance muscle hypertrophy, making them a nutritious and flavorful addition to a muscle-building nutrition plan. 

Mollusks are the second largest category of farmed seafood, accounting for 21% of all global aquaculture production, and oysters represent a key component in the industry [23]. Few studies have shown the potential of marine shellfish for muscle growth [6,24,25,26]. *Crassostrea nippona* is a shellfish that belongs to the class Bivalves, the phylum Mollusks, the order Ostreoida, and the family Ostreidae, and is located in the marine environments of East Asian countries such as Republic of Korea, China, and Japan [27]. In 1913, Fujita T. introduced the taxonomic categorization of *Crassostrea nippona*. A comprehensive study on the taxonomy and phylogeny of Crassostrea species, including *C. nippona*, was conducted by Liu et al. (2017) [28]. Due to the high glycogen component in their bodies, accompanied by a delicious taste and unique flavor, *C. nippona* is preferred in summer over other oyster species, especially the Pacific oysters (*Crassostrea gigas*) [29]. These oyster species contain high-quality proteins along with essential amino acids, vitamins, minerals, and omega-3 fatty acids [30]. Oyster proteins can be degraded into several peptides via enzymatic hydrolysis and have prominent physiological activities. Previous studies have shown that oyster hydrolysates have significant anti-cancer, anti-inflammatory, and antioxidant properties [31]. Nonetheless, the muscle growth effects and anti-muscle atrophy aptitude-related molecular mechanisms of enzyme hydrolysates isolated from shellfish species are not well understood. Hence, further feasibility studies are required to be carried out to explore the mechanism in different species.

Dexamethasone (DEXA) is a synthetic glucocorticoid widely used to induce proteolytic muscle atrophy in both in vivo and in vitro models and used to treat medical conditions such as autoimmune disorders and inflammations [32,33]. It also decreases protein synthesis and promotes the breakdown of proteins related to skeletal muscle mass via the ubiquitin–proteasome systems [34]. DEXA also delivers the necessary environmental conditions to downregulate the mammalian target of the rapamycin mTOR pathway to promote the breakdown of protein synthesis inside the cell [2]. Therefore, it is important to investigate therapeutic approaches to treat this condition to prevent DEXA-induced muscle atrophy, introduce anti-muscle atrophy compounds to prevent important clinical implications, and investigate anti-muscle atrophy drugs or foods to treat these conditions effectively. Numerous studies have shown that enzyme-assisted extraction methods provide outstanding outcomes, including efficient extraction of bioactive compounds with greater extraction yields and protein content, which are highly recognized for superlative muscle growth [6,35,36,37,38,39].

Therefore, this study aimed to prepare oyster hydrolysates from *C. nippona* hydrolyzed with three different enzymes, alcalase, flavourzyme, and protamex, and evaluate their ability to prevent or delay muscle atrophy in DEXA-induced C2C12 cells in vitro. Moreover, a comparative study was also performed to determine which *C. nippona* hydrolysates obtained from alcalase, flavourzyme, and protamex treatments can be applied convincingly for product development in downregulating DEXA-induced muscle atrophy in skeletal muscles of C2C12 cells in vitro. In addition, given that the development of functional foods and the demonstration of the suitability of enzyme hydrolysates of *C. nippona* is also necessary in the modern world, this study determined the effectiveness of these samples in the development of functional foods to treat atrophic disorders, such as sarcopenia, in a promising way.

## 2. Results

### 2.1. General Composition of the Enzyme Hydrolysates of Crassostrea Nippona

The proximate analyses of AOH, FOH, POH, and DWO are summarized in Table 1. Polysaccharides, polyphenols, proteins, lipids, ash, and yield were determined on a dry-weight basis. The yields of AOH, FOH, POH, and DWO were obtained as 75.33 ± 1.08%, 68.33 ± 0.58%, 61.33 ± 0.58%, and 14.00 ± 1.00%, respectively. The yields of all hydrolysates were significantly higher than those of the DWO sample, with that of AOH being the highest. The highest protein content was acquired in AOH as 45.90 ± 2.71%, while FOH, POH, and DWO had protein recovery percentages of 36.02 ± 2.42%, 43.68 ± 1.57%, and 40.05 ± 0.43%, respectively. AOH delivered a higher polysaccharide yield than FOH, POH, and DWO. As expected, the polyphenol content of all four samples was very low, and no significant differences were observed in the lipid profiles of all samples analyzed in this study. 

### 2.2. Cell Cytotoxicity and Proliferation Activity of the Enzyme Hydrolysates of Crassostrea Nippona

The muscle growth activity of the three different enzyme hydrolysates obtained from *C. nippona* was evaluated by assessing cell cytotoxicity, proliferation, and differentiation. The toxic effects and proliferative ability of *C. nippona* in C2C12 cells were determined using CCK8 and BrdU assays, respectively, in a dose-dependent manner. According to the consideration of cell toxicity degrees, results confirmed that all three hydrolysate samples (AOH, FOH, and POH) and DWO were nontoxic to C2C12 cells at 25, 50, and 100 µg/mL concentrations (Figure 1a–d). 

Cell proliferation results showed that both AOH and POH had a significant cell proliferation influence over the DEXA-treated group in C2C12 cells in a dose-dependent manner. FOH did not perform significantly on cell proliferation at the low-concentration ranges of 25 and 50 µg/mL. At the same time, the DWO group was unable to perform any cell-productive effect at any range over the DEXA-treated group. DEXA concentration was maintained constantly throughout the differentiation process at 10 µM [40]. The significance of each hydrolysate sample was assessed over the DEXA treatment group (Figure 1e–h).

### 2.3. Immunostaining and Myotube Diameters on MyHC Expression

Muscle growth was assessed by immunostaining with myosin heavy chain (MyHC) primary antibodies, followed by analysis of the myotube diameters of each stained caption relevant to specific hydrolysate groups and respective concentrations. MyHC observations, nuclear staining of 4′,6-diamidino-2-phenylindole (DAPI), and merged images of MyHC and DAPI for 25, 50, and 100 µg/mL concentrations for AOH and POH are shown in Figure 2a and Figure 2b, respectively. Myotube quantification data for AOH and POH at 25, 50, and 100 µg/mL concentrations is shown in Figure 2c,d. The results indicate that MyHC expression and myotube diameters in differentiated C2C12 cells treated with AOH and POH were significantly upregulated by DEXA in a dose-dependent manner (Figure 2).

### 2.4. Molecular Weight Determination and Amino Acid Composition

The average molecular weight of the AOH-treated differentiated *C. nippona* was determined and is shown in Figure 3. The protein band was visualized as a smear distributed over the 15–75 kDa range and highly concentrated in the 63–75 kDa range. The amino acid compositions of the same sample (AOH) are presented in Table 2.

### 2.5. Protein Expression after Treatment with AOH

Western blot analysis was conducted on the AOH samples to determine the expression of MyHC, myogenin, and MyoD, as shown in Figure 4. The protein expression of MuRF1, Atrogin, Smad 2/3, and Foxo-3a are illustrated in Figure 5. Relative expressions for each were analyzed through ImageJ. MyHC, myogenin, and MyoD were expressively upregulated over the DEXA-treated group, and MuRF-1, Atrogin, p-Smad 2/3, and p-foxo-3a were strongly downregulated in the AOH samples treated with DEXA (10 µM) in a dose-dependent manner. GAPDH was used as an internal reference.

### 2.6. Real-Time Quantitative Polymerase Chain Reaction (RT-qPCR)

Relative mRNA expression levels of myogenin, MyoD, mTOR, MuRF-1, smad 2/3, and foxo-3a were analyzed and are shown in Figure 6. GAPDH was used as an internal reference. In contrast to DEXA, AOH-treated samples showed significantly downregulated MuRF-1, Atrogin, and Foxo-3a mRNA expression and upregulated myogenin, MyoD, and mTOR-related mRNA expression.

## 3. Discussion

Oyster (*C. nippona*) is a famous and commonly consumed shellfish in East Asian countries and has become a popular dietary choice worldwide [23]. Investigating the potential of marine shellfish in muscle growth has also become vital on the industrial scale to develop functional foods and pharmaceutical products [41]. Hence, the current study aimed to investigate the potential of *C. nippona* for human consumption to develop as an impactful functional food for sarcopenia-like muscle atrophic conditions effectively. Therefore, the present study examined the value of AOH on muscle atrophy. Specifically, the authors focused on its potential anti-atrophic effects and explored underlying mechanisms. The overall findings assuredly demonstrate that AOH declined muscle atrophy. Through comprehensive analysis of gene and protein expression levels, the potential of AOH for the regulation of key myokines involved in this process was revealed. Notably, both negative and positive myokines were affected, shedding light on the intricate regulatory network governing muscle preservation. 

Proteins, lipids, and carbohydrates are critical determinants of skeletal muscle growth, with their composition playing a pivotal role in the development of functional foods and dietary supplements. Among these components, proteins are particularly essential for muscle growth and development. Conversely, polysaccharides serve as vital energy sources for cellular metabolism and play a crucial role in hormone modulation and regulation. Notably, the general composition of enzyme hydrolysates revealed significantly higher levels of proteins, lipids, and carbohydrates in AOH, encouraging further investigations into its anti-atrophic activity [41].

Considering both cytotoxicity and cell proliferation, this study shows that the AOH and POH are not toxic to differentiating cells and have significant effects on downregulating muscle atrophy. Based on these criteria, AOH and POH were selected for the immunostaining analysis. The present study reveals that myotube formation was repressed by DEXA (Figure 2). When AOH and POH were subjected to analysis of the formation of myotubes through immunofluorescence techniques, the results confirmed that both AOH and POH were sufficient for their ability to downregulate muscle atrophy in a dose-dependent manner. MyHC was used as a marker to evaluate differentiation. The decline in MyHC expression suggests the incapacitation of myotube formation, which indicates declining muscle regeneration and strength. A similar trend was reported in recent studies of *Ishige okamurae* (IO) and diphloroethohydroxycarmalol (DPHC) against palmitic acid-impaired skeletal myogenesis in C2C12 cells [42,43]. A significant increase in myotube diameters in AOH and POH leads to enhanced myotube fusion of myoblasts to form active mature muscle fibers for treating muscle atrophy [43]. 

Comparative analysis revealed that protein expression was in an optimum state to abate muscle atrophy, detected only in AOH. Therefore, AOH was selected from the entire screening process to conduct this study. The enzyme hydrolysates of *C. nippona* contain a mixture of proteins with a wide range of molecular weights, and SDS-PAGE was used to analyze the molecular weight distribution of the AOH. This aids in our further studies regarding the peptide isolation from AOH depending on the significance of the biological activities. Amino acids also play a vital role in muscle development, and composition analysis of AOH revealed that arginine represented the highest MOL% (15.45%), which is significant for muscle growth, strength, and regeneration. Leucine, isoleucine, and valine are represented as branching amino acids in a considerable range compared to others. These amino acids are highly responsible for the downregulation of muscle breakdown and the assembly of proteins in muscles. Leucine activates mTOR, which is a necessary component of protein synthesis [28]. Isoleucine promotes myogenesis and intramyocellular lipid accumulation, which also facilitates significant protein turnover in the body. Moreover, valine affects the advancement of skeletal tissues, the synthesis of neurotransmitters, energy production, muscle synthesis, and maintenance of immunological parameters effectively [44,45,46]. Additionally, substantial amounts of alanine and glycine are also present in this sample, significantly contributing to muscle growth [47]. Hence, this amino acid profile obtained from AOH was highly beneficial for the development of muscles as well as showing their potential to suppress the atrophic conditions generated in the body due to various stress factors. Alcalase is a serine endopeptidase synthesized from *Bacillus lichenifomis* that has a superior potential for the effective breakdown of proteins from numerous sites, in the formation of a diverse range of peptide fragments, which leads to vital environmental conditions for treating muscle growth and abating muscle atrophy [48]. Therefore, in consideration of the broad view of our future studies regarding peptide isolation and purification, AOH shows high potential for the development of functional foods. 

A Western blotting experiment was performed to evaluate the myokine expression and its regulation by AOH in the DEXA-induced C2C12 cells. Results obtained from Western blotting showed that AOH downregulated muscle atrophic conditions by synchronizing the major constituents contributing to the ubiquitin–proteasome and mTOR signaling pathways. MuRF-1 and Atrogin are E3 ubiquitin ligases involved in the ubiquitin–proteasome system, which leads to the degradation of proteins in the cells [49]. FoxO3a forms a complex with Smad2/3 and activates MuRF-1 transcription. An alternative study revealed that increasing the expression of phosphorylated forms of Smad2/3 and Foxo 3a leads to the upregulation of positive regulators of atrogein [50,51]. Therefore, overexpression of MuRF-1 and Atrogin activated the ubiquitin–proteasome pathway for protein degradation to promote muscle atrophy. The results obtained from Western blot analysis strongly expressed that, under DEXA-induced conditions, AOH has a significant effect on downregulating the ubiquitin ligases of MuRF-1 and Atrogin in a dose-dependent manner (Figure 4). One study reported similar effects by using a marine fish (*Lutjanus guttatus*), and another study detected quercetin-like flavonoids isolated from various plants for muscle atrophy potential [52,53]. MyoD synchronizes the myofibrillar gene expression to facilitate muscle contraction and regeneration. Furthermore, myogenic differentiation is increased by myogenin [54,55]. Due to the atrophy situation, Smad2 and Smad3 targeted MyoD and myogenin and inhibited myoblast and myofiber proliferation and differentiation [2]. The protein expression levels of MyoD and myogenin in the AOH-treated samples were significantly higher than in the DEXA group. Downregulation of the phosphorylated forms of Smad2/3 and Foxo-3a during this study signified an upsurge in the regulation of MyoD and myogenin to mitigate muscle atrophy in AOH-treated C2C12 cells. Hence, this study showed that AOH has a significant impact on abating muscle atrophic conditions and boosts muscle growth by facilitation through myoblast and myofiber proliferation, differentiation, muscle contraction, and regeneration effectively.

The major constituents required for protein synthesis in cells are mTOR [54] and the myosin-heavy chain which contributes to the contractile apparatus in muscle fibers [42]. Hence, protein expressions and mRNA gene expressions showed a significant upregulation of MyHC and mTOR even under DEXA-induced conditions, signifying that AOH can promote sufficient protein and fibers for muscles to avoid atrophic conditions effectively. 

Therefore, the results indicated that AOH has an exceptional effect on attenuating DEXA-induced muscle atrophy in C2C12 cells in an in vitro model by impending negative regulators such as MuRF-1, Atrogin, Smad2/3, and FoxO 3a according to the ubiquitin–proteasome pathway and promoting the expression of positive regulators such as Myogenin, MyoD, mTOR, and MyHC, which correspond to the mTOR signaling pathway (Figure 4 and Figure 5). 

Analogous to the trend of protein expression, the AOH-treated differentiated *C. nippona* significantly downregulated the mRNA expression trend of MuRF-1, Atrogin, and FoxO 3a, and upregulated the expression of Myogenin, MyoD, and mTOR over in DEXA (Figure 5). Downregulation of Foxo-3a also signified that AOH effectively inhibited MuRF-1 and Atrogin in mRNA levels and amplified the gene levels of myogenin and MyoD.

Functional foods are the components that enclose higher nutritional value and maintain superior quality in terms of human health. Tasteful factors, along with the high capacity of safety levels and well-being impacts, are the profound beneficiaries of these products over others [56]. *C. nippona* is a potent oyster shellfish used as an edible portion for human consumption. The outcomes obtained from this study revealed that the AOH of *C. nippona* comprises high nutritional values and works as an outstanding component to fulfill nutrient requirements during the new product development stage. 

Protein expressions at the Act-RIIB and PI3K pathways and their respective receptor level expression analyses were not extensively addressed in this study. Furthermore, previous studies by Gilson et al., 2007, revealed that myostatin contributes a vital role to the formation of stress conditions in the cells followed by the activation of phosphorylated Smad2/3 during muscle atrophic conditions. Hence, further feasibility studies are recommended and encouraged to distinguish the receptor-level mechanism analysis and execute experiments in vivo to confirm the suitability of these hydrolysates for abating muscle atrophy. The isolation of purified peptides will help categorize the molecular mechanism of the active agent of AOH and deliver effective applications in the functional food industry in a meaningful way.

## 4. Materials and Methods

### 4.1. Materials

The chemical constituents and mixtures necessary for all analyses were procured from Sigma-Aldrich (St. Louis, MO, USA). Dulbecco’s Modified Eagle’s medium (DMEM) (Cat. LM 001-05) was used as a growth media for cell culture obtained from WELGENE, Inc. (693, Namcheon-ro, Namcheon-myeon, Gyeongsan-si, Gyeongsangbuk-do 38695, Republic of Korea). C2C12 myoblast cell lines were obtained from the American Type Culture Collection (ATCC; VA, USA). Fetal Bovine Serum (FBS, WELGENE, Inc.). (Cat. S 101-01). and Horse Serum (Cat. 26170-043) (HS, Gibco, Invitrogen Inc.) were two types of serum-supplemented serum supplementing the DMEM. Differentiated C2C12 myotubes were treated with DEXA (Sigma Aldrich, D1756) (CAS-No 50-02-2). Penicillin and streptomycin were treated as antibiotics for the same medium from Invitrogen Inc. (Waltham, MA, USA). Goat anti-mouse IgG H&L (Alexa Fluor TM Plus 555; A32727) for immunofluorescence was obtained from Invitrogen (Cambridge, UK). Primary and secondary antibodies utilized for Western blotting were bought from Santa Cruz Biotechnology (Santa Cruz, CA, USA). Primers for RT-qPCR were purchased from Humanizing Genomics Macrogen, Seoul, Republic of Korea. *Crassostrea nippona* samples were acquired from a local market in Republic of Korea for experimental analysis. Alcalase, protamex, and flavourzyme were the digestive proteases used in this study and were purchased from Novo Co. (Novozyme Nordisk, Bagsvaerd, Denmark).

### 4.2. Preparation of Enzyme Hydrolysate of Crassostrea Nippona

Oysters (*C. nippona*) were purchased from the domestic seafood market in Tongyeong-si, Republic of Korea, from June to August 2022. The oysters’ shells were carefully opened, and the shell was removed. The muscle tissue was then dissected from the remaining body, ensuring minimal inclusion of other tissues. To reduce salinity, the extracted muscle tissue underwent thorough rinsing with running water, followed by an additional rinse using distilled water. Subsequently, the oyster muscle tissue was ground into a fine paste, ensuring uniformity. Finally, the ground tissue was freeze-dried to remove moisture, resulting in powdered oyster samples utilized for further experiments. In this study, powdered oyster samples (20 g) were mixed with 200 mL of distilled water to achieve a 10% substrate concentration. The pH of the mixture was adjusted to 8. Next, three different enzymes—alcalase (7592870), flavourzyme (7612249), and protamex (PW2A1042)—were separately added to the sample, resulting in a final enzyme-substrate concentration of 5.00%. The prepared samples were placed in a shaking incubator (SFDSM06, Jeio Tech, Republic of Korea) and incubated at 55 °C for 24 h with a shaking speed of 130 rpm. Incubated enzyme-hydrolyzed samples were inactivated at 100 °C for 10 min followed by centrifugation at 10,000 rpm for 10 min at 4 °C. Supernatants from each enzymatic hydrolysis sample were collected through vacuum filtration by sourcing a Buchner funnel with Whatman number 4 double filter papers at a time (Cat:1004110, GE Healthcare, Amersham, UK) and triplicates of 1 mL samples were subjected to measure the yield and rest were freeze-dried for 72 h. Freeze-dried samples were collected and allocated to measure the general composition and muscle growth analysis and samples were kept at 4 °C until use for analysis.

### 4.3. General Composition of Crassostrea Nippona 

Polysaccharides, polyphenols, proteins, lipids, ash, and yield content were analyzed in enzyme hydrolysates of oysters in accordance with the AOAC standards [6,57]. The ash content was determined by incinerating the samples in an electric muffle furnace at 555 °C for 5 h (JSMF-30T, JSR Research Inc., 69 Gumsang-gogaegil, Gongju-City, Chungchungnam-Do, Republic of Korea) and the Kjeldahl method (KjeltecTM 8100, FOSS Analytical Slangerupgade 69, Hilleroed, Denmark) was used to determine the protein content. The lipid content was evaluated gravimetrically by the Soxhlet method (FatExtractor E-500, BUCHI Labortechnik AG, Meierseggstrasse 40, Postfach, Flawil, Switzerland). Polysaccharides and polyphenols were measured by the phenol/sulfuric acid method and the Folin–Ciocalteu method, respectively [58,59]. 

### 4.4. Cell Culture and Cell Differentiation

DMEM supplemented with heat-inactivated 10% FBS, 1% penicillin, and streptomycin (Cat. LS 202-02) was used for cell culture experiments and DMEM enriched with 2% HS, (Ref. 26170-043) 1% streptomycin and penicillin was used to differentiate the cultured cells at 37 °C and in a 5% CO_2_ controlled environment (60915672, Sanyo Electric Co., Ltd., Osaka, Japan) [1,2,3,4]. DEXA (CAS-No: 50-02-2) concentration was maintained at 10 µM throughout these experiments to determine whether selected enzyme hydrolysates of *C. nippona* had a significant effect against that in a dose-dependent way [25,60,61]. Initially, the C2C12 cells were seeded into the concentration of 1 × 10^5^ cells/mL in the DMEM medium. Differentiation was initiated once the cells reached 80% confluence, by shifting the medium to DMEM containing 2% HS and 1% PS. The differentiation procedure was followed three times and differentiated cells were subjected to immunostaining, Western blot analysis, and RT-qPCR [62]. For immunostaining initially, cells were seeded into 48-well plates via Western blot and for RT-qPCR we used 5 mL cell culture dishes, respectively. Initially, the total volumes we used for the 48-well plates and 5 mL cell culture dishes were 250 µL and 4 mL, respectively. 

### 4.5. Cytotoxicity Evaluation using Cell Counting Kit-8 (CCK-8 assay)

The cytotoxicity influence of oyster enzyme hydrolysates on the C2C12 cells was achieved using CCK8 assay [63]. C2C12 myoblasts were seeded in a 96-well plate and incubated for 24 h at 37 °C with 5% CO_2_. Viable cell concentration was maintained at 1 × 10^5^ cells/mL throughout the experiment period. Cells were counted by using a LUNAII automated cell counter machine. Here, we used Luna cell counting sides (Cat no: L12001). The concentrations of the enzyme extracts were maintained by considering the cytotoxicity results. The seeded cells were treated with three hydrolysates (AOH, FOH, POH) and DWO with three different concentrations of each extract, from 25 µg/mL, 50 µg/mL, and 100 µg/mL, and incubated for another 24 h at similar environmental conditions. Each well was treated with CCK8 solution and after 2 h the wells were evaluated by taking the absorbance at 450 nm by using a microplate reader (SynergyTM HT, Vermont, USA) [43]. The absence of sample-treated wells was used as a control and calculated as 100% cell viability, and all absorbance readings of sample-treated wells were stated as mean percentage values compared to the untreated wells [64]. 

### 4.6. Cell Proliferation Evaluation using 5-Bromo-2ʹ-deoxyuridine (BrdU) Assay

To estimate the proliferation ability of AHO, FOH, POH, and DWO in C2C12 cells, the BrdU assay (Millipore, Billerica, MA, USA) was directed by handling Bromodeoxyuridine (BrdU) labeling and a detection kit (11647229001, RocheR Life Science Products, Penzberg, Upper Bavaria, Germany). Seeded C2C12 cells were treated with digested AHO, FOH, POH, and DWO during the cell differentiation period under a differentiation medium (DMEM + 2% HS + 1% PS). The proliferation capability of the cells treated with digested samples was compared with that of the DEXA-treated sample and DEXA was evaluated over control sample wells. The background effect of the media was eliminated by acquiring the absorbance by using the wells that contained only the media without any serum and cells. The absorbance of the samples and background were measured in an ELISA reader at 370 nm [62]. 

### 4.7. Immunostaining of Myosin Heavy Chain (MyHC)

Differentiated myotubes were washed three times with cold 1× PBS (20× PBS consisted of 2.7 M sodium chloride (NaCl), 54 mM potassium chloride (KCl), 86 mM sodium phosphate dibasic (Na_2_HPO_4_), and 28 mM potassium phosphate monobasic (KH_2_PO_4_), all dissolved in distilled water. The final pH was maintained at 7.2–7.5 and fixed with chilled methanol for 5 s. Fixed myotubes were washed three times with PBST (PBS + 1% Tween 20). Fixed and permeabilized cells were blocked with 1% BSA (Cat No. BSAS 0.1) containing 22.52 mg/mL glycine (CAS No. 56-40-6) and incubated with MHC-specific primary antibodies (Santa Cruz) (1:250 in 1% BSA) for 2 h at room temperature. After being washed three times with PBST, incubated cells were labeled with goat anti-mouse IgG H&L (Alexa Fluor TM Plus 555; A32727)-conjugated secondary antibodies (Santa Cruz) (1:1000 in 1% BSA) for 2 h at room temperature, followed by a similar washing step before nuclear staining. Cell nuclei were stained with 300 nm DAPI (40,6-diamidino-2-phenylindole; Sigma-Aldrich), and the final washing step was performed using PBST. Finally, fluorescent images of the stained myotubes were visualized using a confocal microscope (Carl Zeiss, Oberkochen, Germany) [44,65]. Myotube diameters were calculated for control, DEXA, and DEXA-treated samples of AOH and POH using ImageJ software 1.49v (National Institutes of Health, Bethesda, MD, USA) according to the confocal microscope captions referring to each well, and significance was calculated for each sample over the DEXA-treated control value [62]. 

### 4.8. Molecular Weight Determination and Amino Acid Composition

After confirmation of cytotoxicity, cell proliferation ability, and myotube diameter, AOH was selected for Western blotting and RT-qPCR analysis. Molecular weight determination was conducted using sodium dodecyl sulfate–polyacrylamide gel electrophoresis (SDS-PAGE) on 4% stacking and 12% separating polyacrylamide gels, and amino acid composition of the chosen sample was performed using Agilent 1260 series HPLC conditions [66,67]. An AOH sample (100 mg/mL) was subjected to SDS PAGE [45]. Waters Nova-Pak C18 4 μm (3.9 × 300 mm) was the column used during the HPLC analysis. The two mobile phases applied in this analysis were 140 mM Sodium Acetate trihydrate, 0.15% TEA, 0.03% EDTA, 6% CH3CN, pH 6.1, and 60% CH3CN, 0.015% EDTA. The flow rate, run time, and equilibration time were 1.0 mL/min, 30 min, and 10 min, respectively. The HP 1100 Series at 254 nm was used to detect the peaks. AOH (100 mg/mL) was used for the SDS-PAGE analysis [68]. 

### 4.9. Western Blot Analysis

The digested AOH-treated cells were harvested using PBS (Cat. PR2007-100-00)/trypsin (Cat LS 015-01) and lysed (RIPA Buffer (R4100-050). The total protein content (TPC) was calculated by BCA assay (23227, Thermo Fisher Scientific), and an equivalent amount of denatured protein (30 µg) was loaded into each well. The separated proteins were blotted onto a nitrocellulose membrane (Amersham™ ProtranTM 0.45 µm NC). A total of 5% skim milk in Tris-buffered saline (TBS) (0.05% Tween-20) was applied to block the protein-containing membrane for 2 h at room temperature. The blocked membrane was incubated overnight with primary antibodies (Myosin heavy chain, MyoD, Myogenin, phosphorylated Forkhead box protein (p-Foxo-3a), Forkhead box protein (Foxo-3a), Muscle RING-finger protein-1 (MuRF-1), and atrogin protein) which were diluted to 1:1000 in 5% skim milk in 1% TBS with 0.1% Tween 20 detergent (TBST) at 4 °C for 16 h followed by augmented fluorescent tagged respective mouse or rabbit secondary antibodies (1:3000) into the primary antibody-tagged proteins. The original concentration of TBS was 20× (20× TBS contains 160 g of NaCl, 4 g of KCl, and 60 g of Tris in 1000 mL of distilled water; the final pH was 7.6). Glyceraldehyde 3-phosphate (GAPDH) was used as a housekeeping protein during the analysis of samples using different antibodies. The membranes were imaged using a chemiluminescent substrate (Amersham, Arlington Heights, IL, USA) and quantified using ImageJ 1.49v software (National Institutes of Health, Bethesda, MD, USA) [63]. 

### 4.10. RT-qPCR

The selected AOH samples at 25, 50, and 100 µg/mL were differentiated as previously mentioned in Section 2.4. The cells were then harvested, and total Ribonucleic Acid (RNA) was extracted using TRIzol reagent (Invitrogen, CA, USA) following the manufacturer’s instructions. RNA content was measured using a µDrop plate (Thermo Scientific, Waltham, MA, USA), and 2 µg of RNA was reverse-transcribed into cDNA using a first-strand cDNA synthesis kit (TaKaRa, Shiga, Japan). The primers used in the experiment are listed in Table 3. The PCR cycling conditions were as described by [65]. Here, initial denaturation was performed at 95 °C/30 s. Annealing and extension were conducted in 30 cycles (95 °C/5 s, 55 °C/10 s, and 72 °C/20 s. Final extension 95 °C/15 s, and 60 °C/30 s). Livak and Schmittgen (2001) implemented a scheme to quantify the relative gene expression [68]. GAPDH was used as an internal control to normalize the relative gene expression levels.

### 4.11. Statistical Analysis

To realize the final results, all the analyses were performed in triplicate. Data are expressed as means ± standard deviation. Statistical significance was evaluated using one-way analysis of variance (ANOVA) and Tukey’s test. Two-tailed Student’s *t*-tests were performed when two conditions were required for the analysis. The significance is denoted compared to the control; *p* < 0.0001 “####” and over DEXA level was defined; *p* < 0.05 “*”, *p* < 0.01 “**”, *p* < 0.001 “***”, and *p* < 0.0001 “****”.

## 5. Conclusions

In conclusion, this study robustly supports the anti-muscle atrophic effect of AOH derived from *C. nippona*. Through a comprehensive analysis of gene and protein expression levels, AOH downregulates negative regulators (MuRF-1, Atrogin-1, Smad2/3, and FoxO 3a) associated with the ubiquitin–proteasome pathway. Simultaneously, AOH promotes the expression of positive regulators (Myogenin, MyoD, mTOR, and MHC) related to the mTOR signaling pathway. The study outcomes confirmed the muscle growth and anti-atrophy effects of AOH in a DEXA-induced muscle atrophy model using C2C12 cells. Amino acid composition and cell-protective aptitude analyses further highlight AOH’s potential for treating muscle atrophy effectively compared to other hydrolysates. Receptor-level analysis and in vivo experiments are recommended to strengthen these findings and facilitate impactful industrial product development for muscle atrophy treatment.

## Figures and Tables

**Figure 1 marinedrugs-22-00113-f001:**
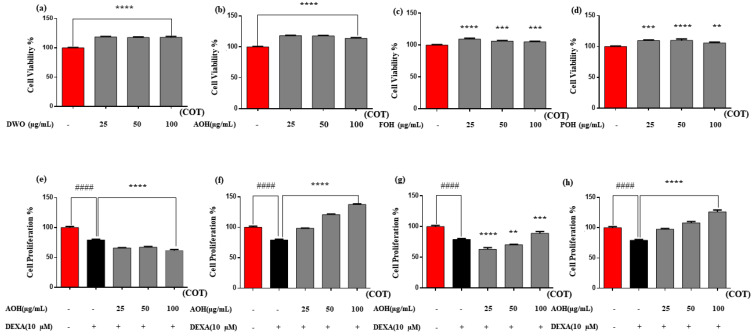
Cell viability % of enzyme hydrolysates of *C. nippona*. (**a**) DWO; (**b**) AOH; (**c**) FOH; (**d**) POH; cell proliferation effect of protein hydrolysate of *C. nippona* (**e**) DWO; (**f**) AOH; (**g**) FOH; (**h**) POH. Experiments were carried out in triplicate and the results are presented as means ± SD (n = 3). Significance is denoted compared to the control #### *p* < 0.0001 and DEXA groups; ** *p* < 0.01, *** *p* < 0.001, **** *p* < 0.0001. DWO: distilled water oyster; AOH: alcalase oyster hydrolysate; FOH: flavourzyme oyster hydrolysate; POH: protamex oyster hydrolysate; DEXA: dexamethasone; COT: concentration of treatments.

**Figure 2 marinedrugs-22-00113-f002:**
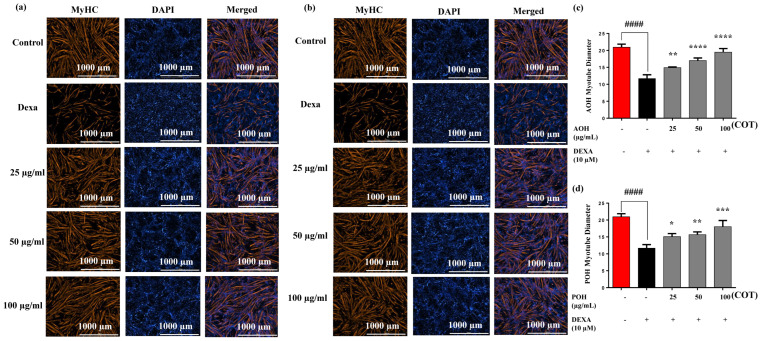
Immunofluorescence analysis of myosin heavy chains of enzyme hydrolysates of *Crassostrea nippona*. (**a**) AOH; (**b**) POH; (**c**) myotube diameter scoring analysis for AOH; (**d**) POH. Experiments were carried out in triplicate and the results are presented as means ± SD (n = 3). Significance is denoted compared to the control #### *p* < 0.0001 and DEXA group; ** *p* < 0.01, *** *p* < 0.001, **** *p* < 0.0001. DWO: distilled water oyster; AOH: alcalase oyster hydrolysate; POH: protamex oyster hydrolysate; DEXA: dexamethasone; COT: concentrations of treatments.

**Figure 3 marinedrugs-22-00113-f003:**
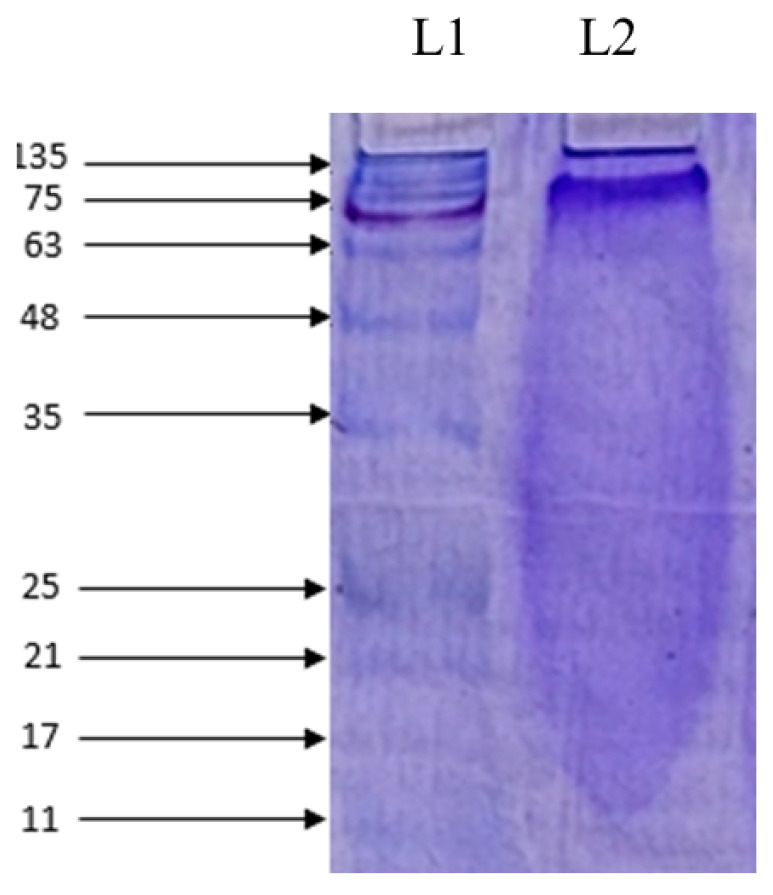
Determination of the molecular weight distribution of AOH by SDS-PAGE. Lane 1 (L1): protein marker (5–245 kDa) and Lane 2 (L2): alcalase oyster hydrolysate (AOH).

**Figure 4 marinedrugs-22-00113-f004:**
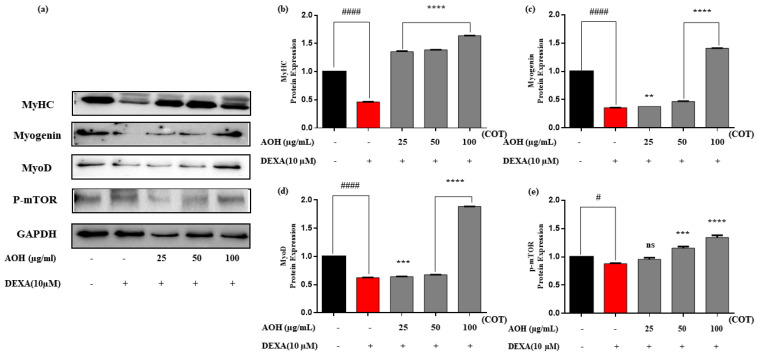
Evaluation of myogenic markers in differentiating myoblasts through myotubes in AOH of *C. nippona*. (**a**) Western blotting analysis, relative expression of (**b**) MyHC, (**c**) myogenin, (**d**) MyoD, (**e**) p-mTOR. Significance is denoted compared to control # *p* < 0.05, #### *p* < 0.0001 and compared to DEXA group ns *p* > 0.05, ** *p* < 0.01, *** *p* < 0.001, and **** *p* < 0.0001. AOH: alcalase oyster hydrolysate; MyHC: myosin heavy chain; mTOR: mammalian target of rapamycin; DEXA: dexamethasone; COT: concentration of treatments.

**Figure 5 marinedrugs-22-00113-f005:**
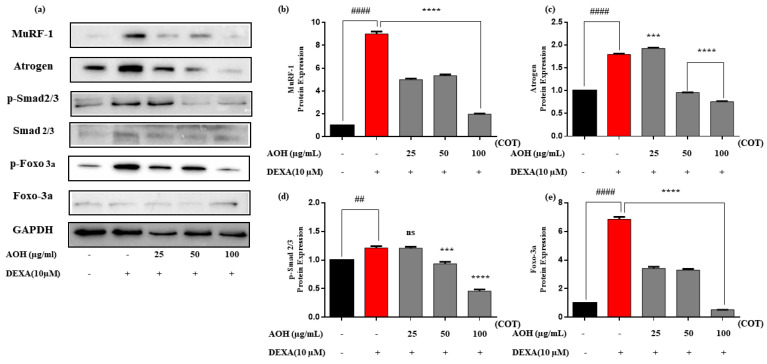
Evaluation of myogenic markers in differentiating myoblasts through myotubes in AOH of *C. nippona*. (**a**) Western blotting analysis, relative expression of (**b**) MuRF-1, (**c**) Atrogin, (**d**) Smad2/3, and (**e**) Foxo-3a. Significance is denoted compared to control ## *p* < 0.01, #### *p* < 0.0001 and compared to DEXA group ns *p* > 0.05, *** *p* < 0.001, and **** *p* < 0.0001. AOH: Alcalase oyster hydrolysate; MuRF-1: Muscle RING-finger protein-1; Foxo: Forkhead box protein; DEXA: dexamethasone; COT: concentration of treatments.

**Figure 6 marinedrugs-22-00113-f006:**
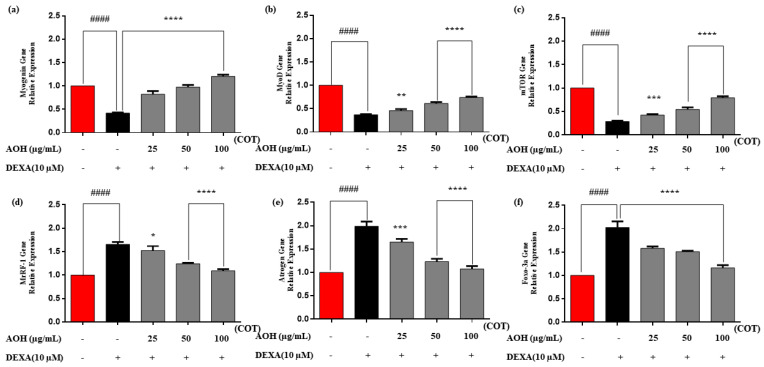
Gene expression of mRNA levels via RT qPCR analysis in AOH of *C. nippona*; (**a**) Myogenin, (**b**) MyoD, (**c**) mTOR, (**d**) MuRF-1), (**e**) Atrogin, (**f**) Foxo 3a. The data are represented as mean ± SE. Significance is denoted compared to control #### *p* < 0.0001 and compared to DEXA group * *p* < 0.05, ** *p* < 0.01, *** *p* < 0.001, and **** *p* < 0.0001. mTOR: mammalian target of rapamycin; MuRF-1: Muscle RING-finger protein-1; Foxo: Forkhead box protein; DEXA: dexamethasone; COT: concentration of treatments.

**Table 1 marinedrugs-22-00113-t001:** Hydrolysate yield and proximate analysis of the protein hydrolysate of *Crassostrea nippona* on muscle growth.

Component	Composition			
	DWO	AOH	FOH	POH
Polysaccharide	2.72 ± 0.47%	23.83 ± 1.42% ****	6.86 ± 0.45% ****	11.83 ± 0.50% ****
Polyphenol	0.47 ± 0.01%	0.51 ± 0.00% ****	0.45 ± 0.01%	0.50 ± 0.01% **
Proteins	40.05 ± 0.43%	45.90 ± 2.71% ****	36.02 ± 2.42%	43.68 ± 1.57% ****
Lipids	6.54 ± 1.56%	8.94 ± 1.67% ****	8.14 ± 1.42% ****	8.50 ± 1.22% ****
Ash	14.00 ± 1.00%	75.33 ± 1.08% ****	68.33 ± 0.58% ****	61.33 ± 0.58% ****
Yield	0.47 ± 0.01%	0.51 ± 0.00% ****	0.45 ± 0.01%	0.50 ± 0.01% **

The experiments were triplicated and data are represented as mean ± SE. Significance is denoted compared to control. ** *p* < 0.01., **** *p* < 0.0001; DWO: distilled water oyster; AOH: alcalase oyster hydrolysate; FOH: flavourzyme oyster hydrolysate; POH: protamex oyster hydrolysate.

**Table 2 marinedrugs-22-00113-t002:** Amino acid composition of AOH of *Crassostrea nippona*.

Amino Acid	Moles Percentage (%)	g/100 g
Cysteine and cystine	0.39%	0.223
Asparagine and asparatic acid	8.36%	3.429
Glutamine and glutamic acid	11.02%	4.997
Serine	4.62%	1.503
Glycine	10.21%	2.361
Histidine	1.50%	0.715
Arginine	15.43%	8.284
Threonine	4.66%	1.709
Alanine	9.70%	2.665
Proline	4.17%	1.478
Tyrosine	1.87%	1.046
Valine	5.46%	1.972
Methionine	2.19%	1.008
Isoleucine	4.65%	1.881
Leucine	6.95%	2.809
Phenylalanine	3.06%	1.558
Tryptophan	0.00%	0.000
Lysine	6.12%	2.758
Total	100.00%	40.173

**Table 3 marinedrugs-22-00113-t003:** Sequence of primers used in this study.

Gene	Primer	Sequence
GAPDH	Sense Antisense	5′-AAGGGTCATCATCTCTGCCC-3′ 5′-CCACGATGGACGTAAGGGAG-3′
MyoD	Sense Antisense	5′-GCCGCCTGAGCAAAGTGAATG-3′ 5′-CAGCGGTCCAGTGCGTAGAAG3′
Myogenin	Sense Antisense	5′-GTCCCAACCCAGGAGATCAT-3′ 5′-CCACGATGGACGTAAGGGAG-3′
mTOR	Sense Antisense	5′-CACATCACTCCCTTCACCA-3′ 5′-GCAACAACGGCTTTCCAC-3′
MuRF-1	Sense Antisense	5′-ATCTAGCCTGATTCCTGATGGA-3′ 5′ACCACAGGCTTGGTAAACATCT3′
Foxo-3a	Sense Antisense	5ACCTTCGTCTCTGAACCTTG–3′ 5′AGTGTGACACGGAAGAGAAGGT3′
Smad 2/3	Sense Antisense	5GTCCCAACCCAGGAGATCAT–3′ 5′CCACGATGGACGTAAGGGAG3′

## Data Availability

All data are contained within the manuscript.

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
