# Peer review of "Exploring the Potential of Crassostrea nippona Hydrolysates as Dietary Supplements for Mitigating Dexamethasone-Induced Muscle Atrophy in C2C12 Cells"

_marinedrugs, 2024, doi:10.3390/md22030113_

Round 1

Reviewer 1 Report

Comments and Suggestions for Authors

Muscle atrophy is a detrimental and injurious condition that leads to reduced skeletal muscle mass and disruption of protein metabolism. Based on the truth that Marine organisms are abundant sources of high-quality proteins, essential amino acids, and bioactive compounds, which are fundamental for muscle growth and repair. And the omega-3 fatty acids found in certain shellfish varieties further contribute to muscle health by reducing inflammation and supporting overall muscle function, authors tried to evaluate the oyster hydrolysates hydrolyzed with three different enzymes (AOH, FOH, POH) and DWO on DEXA-induced muscle atrophy in C2C12 cells. Good story. However, I believe that the story can be further improved by reducing the commonsense text. Also, there are some suggestions that may be useful for the improvement of paper.

(1)   Abstract, move the 5th sentence to the first, since the scientific problems is the most important thing that readers want to know first.

(2)   Introduction, move the 3rd paragraph to the first, merge the first three paragraphs together. Merge the 4 and 5th paragraphs together too.

(3)   Results, why did the authors not draw a schematic representation for the mitigating mechanism of AOH on DEXA-induced muscle atrophy, it is helpful for readersto understand of mechanism of action.

(4)   Results, provide rulers for all Immunofluorescence images.

(5)   Results, I can’t find the data of POH, FOH and DWO from 2.4 to 2.6, maybe DWO is a good negative control for the interpretation mechanism.

Comments on the Quality of English Language

Minor editing of English language required

Reviewer 2 Report

Comments and Suggestions for Authors

This study has some potential scientific interest, but it is too preliminary. The authors used only one cell line. This is inadequate. Several cell lines should be used. Furthermore, to study “dietary supplementation” according to the title of the manuscript, one should use some in vivo models. PCR-based analysis of mRNA levels must be supplemented with the analysis of protein levels and in the case of kinase activity, e.g. mTOR, the activity (the phosphorylation status) must be documented. For FOXO3a, please also check the changes in cellular localization. In conclusion, study design is inadequate.

Data presentation: there are issues with limited figure readability (the labels are too small). Furthermore, microphotographs are hard to read.

Round 2

Reviewer 1 Report

Comments and Suggestions for Authors

The authors properly revised the manuscript, I suggest that this manuscript can be accepted for publication now. 

Author Response

Thank you for your commitment. Please find the attachments for your kind reference

Reviewer 2 Report

Comments and Suggestions for Authors

The manuscript was not adequately improved according to my comments. The authors have ignored my comments. Please note that it is a standard to use several cell lines to document a cellular phenomenon and related mechanisms. Thus, the study is still too preliminary (one cell line-based) to be published in a scientific journal. The authors are convinced that one can study “dietary supplementation” using a cell line model in vitro. “Dietary supplementation” must be studied using in vivo models. To study the activation of kinases such as mTOR both the levels of unphosphorylated and phosphorylated form of protein kinase must be analyzed (the activation is calculated then as a ratio of phosphorylated form to unphosphorylated form). To conclude, the manuscript was not amended according to my comments. Rejection is recommended without any further revision.

Author Response

Thank you for your valuable consideration. Please find the attached files for your kind reference.
